# PD-L1 and AKT Overexpressing Adipose-Derived Mesenchymal Stem Cells Enhance Myocardial Protection by Upregulating CD25^+^ T Cells in Acute Myocardial Infarction Rat Model

**DOI:** 10.3390/ijms25010134

**Published:** 2023-12-21

**Authors:** Yu-Kai Lin, Lien-Cheng Hsiao, Mei-Yao Wu, Yun-Fang Chen, Yen-Nien Lin, Chia-Ming Chang, Wei-Hsin Chung, Ke-Wei Chen, Chiung-Ray Lu, Wei-Yu Chen, Shih-Sheng Chang, Woei-Cheang Shyu, An-Sheng Lee, Chu-Huang Chen, Long-Bin Jeng, Kuan-Cheng Chang

**Affiliations:** 1Division of Cardiovascular Medicine, China Medical University Hospital, Taichung 404327, Taiwanthinkingandwriting@gmail.com (Y.-N.L.); motic721@gmail.com (W.-H.C.); kennychen0205@gmail.com (K.-W.C.);; 2Cardiovascular Research Laboratory, China Medical University Hospital, Taichung 404327, Taiwan; freedsun1@yahoo.com.tw (C.-M.C.); anshenglee33@gmail.com (A.-S.L.); 3School of Medicine, China Medical University, Taichung 404328, Taiwan; 4School of Post-Baccalaureate Chinese Medicine, China Medical University, Taichung 404328, Taiwan; meiyaowu0919@gmail.com; 5Department of Chinese Medicine, China Medical University Hospital, Taichung 404327, Taiwan; 6Department of Medicine, Mackay Medical College, New Taipei City 25245, Taiwan; sgarraache@hotmail.com (Y.-F.C.); s40904@hotmail.com (W.-Y.C.); 7Graduate Institute of Biomedical Sciences, China Medical University, Taichung 404328, Taiwan; shyu9423@gmail.com; 8Translational Medicine Research Center, China Medical University Hospital, Taichung 404327, Taiwan; 9Neuroscience and Brain Disease Center, New Drug Development Center, China Medical University, Taichung 404328, Taiwan; 10Department of Neurology, China Medical University, Taichung 404328, Taiwan; 11Department of Occupational Therapy, Asia University, Taichung 413305, Taiwan; 12Vascular and Medicinal Research, Texas Heart Institute, Houston, TX 77030, USA; mendelchen@hearthrtllc.com; 13New York Heart Research Foundation, Mineola, NY 11514, USA; 14Cell Therapy Center, China Medical University Hospital, Taichung 404327, Taiwan; longbin.cmuh@gmail.com; 15Organ Transplantation Center, China Medical University Hospital, Taichung 404327, Taiwan

**Keywords:** adipose-derived mesenchymal stem cells (AdMSC), myocardial infarction, programmed death ligand 1 (PD-L1), regulatory T cells

## Abstract

This study explores the synergistic impact of Programmed Death Ligand 1 (PD-L1) and Protein Kinase B (Akt) overexpression in adipose-derived mesenchymal stem cells (AdMSCs) for ameliorating cardiac dysfunction after myocardial infarction (MI). Post-MI adult Wistar rats were allocated into four groups: sham, MI, ADMSC treatment, and ADMSCs overexpressed with PD-L1 and Akt (AdMSC-PDL1-Akt) treatment. MI was induced via left anterior descending coronary artery ligation, followed by intramyocardial AdMSC injections. Over four weeks, cardiac functionality and structural integrity were assessed using pressure–volume analysis, infarct size measurement, and immunohistochemistry. AdMSC-PDL1-Akt exhibited enhanced resistance to reactive oxygen species (ROS) in vitro and ameliorated MI-induced contractile dysfunction in vivo by improving the end-systolic pressure–volume relationship and preload-recruitable stroke work, together with attenuating infarct size. Molecular analyses revealed substantial mitigation in caspase3 and nuclear factor-κB upregulation in MI hearts within the AdMSC-PDL1-Akt group. Mechanistically, AdMSC-PDL1-Akt fostered the differentiation of normal T cells into CD25^+^ regulatory T cells in vitro, aligning with in vivo upregulation of CD25 in AdMSC-PDL1-Akt-treated rats. Collectively, PD-L1 and Akt overexpression in AdMSCs bolsters resistance to ROS-mediated apoptosis in vitro and enhances myocardial protective efficacy against MI-induced dysfunction, potentially via T-cell modulation, underscoring a promising therapeutic strategy for myocardial ischemic injuries.

## 1. Introduction

Diminished myocardial perfusion, culminating in cardiomyocyte death, serves as a principal determinant of morbidity and mortality following acute myocardial infarction (AMI) [1,2]. Given that the regenerative capacity of adult cardiomyocytes is generally regarded as inadequate—attributed to limited proliferative abilities and a scarcity of endogenous cardiac stem cells—stem cell transplantation has emerged as a promising therapeutic avenue for post-AMI cardiac repair [2,3,4,5]. Techniques involving the direct intramyocardial injection of stem cell suspensions, as well as intracoronary delivery methods, are commonly employed in the context of AMI [6,7,8]. Evidence from both basic and applied research supports the efficacy of stem cell-based therapies in ameliorating AMI-induced cardiac dysfunction [2,9,10,11,12]. Various stem cell types originating from adult tissues, including hematopoietic stem cells, endothelial progenitor cells, cardiac progenitor cells, skeletal myoblasts, bone marrow-derived mesenchymal stem cells (BMMSCs), and adipose-derived mesenchymal stem cells (AdMSCs), have been explored for their potential in cardiac and vascular repair [2,4,13]. Nevertheless, unresolved debates persist concerning the optimal cell type, dosage, administration route, and timing for transplantation [2].

Over recent years, mesenchymal stem cells (MSCs) have been demonstrated to possess both anti-inflammatory and immunomodulatory properties [14,15]. Their therapeutic potential has been corroborated by findings from earlier clinical trials and contemporary meta-analyses [16,17,18]. Beyond conventional cell transplantation methods, tissue-engineered pericardial patches incorporating autologous AdMSCs have shown efficacy in treating myocardial infarction (MI) in a rabbit model [19]. Another study reported significant advantages in preserving left ventricular function and mitigating post-AMI remodeling when AdMSCs were incorporated into platelet-rich fibrin scaffolds in rat models [20]. Among the various MSC types, AdMSCs are distinguishable for their ease of isolation from adipose tissue harvested from multiple anatomical sites, which contain a higher yield of MSCs [21]. Moreover, AdMSCs can be procured using minimally invasive techniques, further favoring their application [22,23].

It is crucial to acknowledge that the post-MI infarct zone presents a hostile microenvironment characterized by hypoxia, acidosis, the accumulation of cytotoxic waste, and nutrient scarcity, which collectively compromise the survival and engraftment of transplanted stem cells in the damaged myocardium [2,24]. Various pre-treatment strategies have been investigated to ameliorate this issue. These include the upregulation of Protein kinase B (Akt) [25,26], the application of endothelial nitric oxide synthase (eNOS)-enhancing agents [27], hypoxic preconditioning [28], Angiotensin II exposure [29], as well as preconditioning with stromal cell-derived factor 1 alpha (SDF1α) and vascular endothelial growth factor (VEGF) overexpression [30]. These interventions have shown efficacy in enhancing cardiac function in the context of post-infarct cardiomyopathy, primarily by improving the survival rates of the implanted cells [2].

It is widely accepted that cardiovascular diseases are modulated by immunological processes, with immune responses playing a significant role in exacerbating the severity of atherosclerosis. T cells are implicated in every stage of atherosclerotic progression, and attenuating hyperactive T cell activity is anticipated to ameliorate inflammatory conditions [31]. Numerous investigations have targeted the inhibition of immune responses within the intimal layer or atherosclerotic plaques through the suppression of T cell activities. The coinhibitory pathway involving Programmed Death Receptor 1 (PD-1) and Programmed Death Ligand 1 (PD-L1) serves a critical function in the establishment of immune tolerance by mitigating unwarranted immune cell activation.

For further elucidation of stem cells’ potential in the field of regenerative medicine, cellular modifications may be requisite. In the current investigation, we employed genetic engineering techniques to induce the overexpression of both PD-L1 and Akt in AdMSCs. This modification aims to temper excessive inflammation and immunological responses. Additionally, we evaluated the postulation that the simultaneous activation of these signaling pathways would generate a synergistic effect, thereby offering enhanced therapeutic efficacy for the treatment of AMI in a rat model.

## 2. Results

### 2.1. Characterization of AdMSCs Overexpressing PD-L1 and Akt

To investigate the functional overexpression of Akt and PD-L1 proteins in AdMSCs post-transfection, both immunofluorescence assays and Western blot analyses were conducted. Immunofluorescence findings revealed an augmented abundance of both PD-L1 and Akt proteins in AdMSCs transduced with pUltra-PD-L1-Akt compared to non-transduced AdMSCs (Figure 1A). Furthermore, Western blot quantification demonstrated a statistically significant upregulation of PD-L1 and Akt proteins in the transfected AdMSCs (PD-L1: 83.09 ± 4.88% relative to GAPDH; Akt: 99.65 ± 21.13% relative to GAPDH) in comparison to control AdMSCs (PD-L1: 1.84 ± 0.94% relative to GAPDH; Akt: 31.80 ± 7.48% relative to GAPDH) (Figure 1B,C). Collectively, these data substantiate that the transfection process utilizing pUltra-PD-L1-Akt successfully and stably upregulated the expression levels of PD-L1 and Akt in AdMSCs. These modified AdMSCs were subsequently utilized to explore their potential therapeutic effects on MI.

Cell proliferation rates in AdMSC and AdMSC-PD-L1-Akt were assessed using Carboxyfluorescein Succinimidyl Ester (CFSE) assays and quantified via flow cytometry. ADMSCs exhibited an approximate proliferation rate of 80% (Figure 2A). Notably, this rate was significantly enhanced in AdMSC-PD-L1-Akt cells, registering at 92.80 ± 3.64% compared to 79.86 ± 4.78% in AdMSCs. Furthermore, the oxidative stress resistance of both cell types was evaluated by assessing their viability post 24 h exposure to varying concentrations of hydrogen peroxide (H_2_O_2_). At a concentration of 10 μM H_2_O_2_, cell viability in AdMSCs was reduced to approximately 80%, while AdMSC-PD-L1-Akt cells remained substantially more resistant to such oxidative injury (Figure 2B; 93.31 ± 1.31% vs. 82.64 ± 1.59% at 10 μM H_2_O_2_). Given that intracellular ROS levels serve as vital indicators of oxidative stress, we further probed differences in H_2_O_2_-induced ROS generation between the two cell types utilizing the H2DCFDA assay. Exposure to 10 μM H_2_O_2_ led to a time-dependent increase in ROS levels in AdMSCs, reaching approximately 30% of baseline values at 120 min. In contrast, AdMSC-PD-L1-Akt exhibited a more moderate increase in ROS levels, with only a 15% rise recorded following the same hydrogen peroxide treatment (Figure 2C; 129.33 ± 0.70% vs. 118.37 ± 1.48% at 120 min).

### 2.2. PD-L1 and AKT-Overexpressing AdMSCs on Post-MI Cardiac Dysfunction and Infarct Size

To evaluate the therapeutic efficacy of AdMSCs engineered to overexpress PD-L1 and AKT (AdMSC-PD-L1-Akt) in attenuating MI-induced cardiac dysfunction, these cells were administered to rats immediately post-ligation of the left anterior descending artery (LAD). The localization of AdMSC-PD-L1-Akt at the cardiac injection site after 24 h was confirmed using in vivo imaging (Appendix A). Representative pressure–volume (P-V) loop results indicated alterations in cardiac function across all experimental groups (Figure 3A). Specifically, a notable decline in the slope of the end-systolic pressure–volume relationship (ESPVR) was observed in the MI group (0.82 ± 0.08 mmHg/μL) compared to the sham controls (1.61 ± 0.18 mmHg/μL). This reduction was mitigated in the MI+AdMSC-PD-L1-Akt group but not in the MI+AdMSC group (Figure 3B; 1.26 ± 0.22 and 0.97 ± 0.12 mmHg/μL, respectively). Although a slight elevation was seen in the slope of the end-diastolic pressure–volume relationship (EDPVR) in the MI group (0.010 ± 0.005 mmHg/μL) relative to sham controls (0.005 ± 0.002 mmHg/μL), this difference was not statistically significant across the groups (Figure 3C). Further analysis indicated that preload-recruitable stroke work (PRSW), a parameter indicative of systolic function, also exhibited a decline in MI-affected animals (18.35 ± 2.16 mmHg × μL) compared to the sham group (29.83 ± 3.19 mmHg × μL). This decline was ameliorated in the MI+AdMSC-PD-L1-Akt group but not in the MI+AdMSC cohort (Figure 3D; 30.57 ± 4.05 and 21.35 ± 3.06 mmHg × μL, respectively).

In addition to evaluating cardiac function, we assessed the potential of AdMSC-PDL1-Akt to mitigate MI-induced cardiac tissue damage. Infarct size and the area at risk (AAR) were delineated in four distinct animal groups using 2,3,5-triphenyltetrazolium chloride (TTC; red) and phthalocyanine blue staining (blue), as illustrated in Figure 4A. Comparative analysis of the AAR across the three groups revealed no statistically significant differences, suggesting uniformity in the ischemic extents across all cohorts (Figure 4B). Relative to the MI-only group, which displayed an infarct size of 64.24 ± 8.49%, the MI+AdMSC cohort showed a modest reduction with an infarct size of 40.37 ± 11.12%. Notably, the MI+AdMSC-PDL1-Akt group demonstrated a substantially diminished infarct size, recorded at 17.65 ± 5.30% (Figure 4C).

### 2.3. PD-L1 and AKT-Overexpressing AdMSCs on Apoptotic/Inflammatory Markers and Immune Regulation

To elucidate the protective mechanism exerted by AdMSC-PDL1-Akt against MI, we conducted an in-depth evaluation of several inflammatory biomarkers, namely, caspase3, NFκB, and CD25, within cardiac tissues using immunohistochemistry techniques (Figure 5A). Notably, MI augmented the expression levels of both caspase3 and NFκB, suggestive of MI-induced apoptosis and inflammation, respectively (Figure 5B; 13.40 ± 1.04% vs. 0.94 ± 0.07% for caspase3 and 14.23 ± 2.84% vs. 0.89 ± 0.31% for NFκB). This elevation was significantly attenuated in the MI+AdMSC and MI+AdMSC-PDL1-Akt cohorts, with values of 6.19 ± 1.19% and 2.62 ± 0.45% for caspase3 and 6.15 ± 0.87% and 2.89 ± 0.58% for NFκB, respectively. Conversely, both AdMSC and AdMSC-PDL1-Akt markedly enhanced CD25 abundance, registering at 0.12 ± 0.09% and 0.32 ± 0.10%, respectively. In comparison, the MI group did not exhibit significant variations in CD25 levels relative to the sham group, with a measured value of 0.01 ± 0.01%.

Given the pronounced upregulation of CD25 expression observed in the MI+AdMSC-PDL1-Akt group in vivo, we sought to elucidate the influence of AdMSC-PDL1-Akt on T cell activity. To ascertain whether AdMSC-PDL1-Akt could potentiate the expression of CD4 ^+^ CD25^+^ T cells in vitro, we isolated murine CD4^+^ T cells and co-cultivated them either with standard AdMSC or AdMSC-PDL1-Akt. The progression of T cell differentiation within the AdMSC-mediated co-culture milieu was monitored over a span of 4 days using flow cytometry (Figure 6A). By day 4, there was a modest elevation in the proportions of CD4^+^CD25^+^ T cells within the AdMSC co-culture system relative to controls (Figure 6B; 1.41 ± 0.15% vs. 0.97 ± 0.12%; *p* = 0.033). Strikingly, the CD25 expression was more robustly augmented in the AdMSC-PDL1-Akt co-culture setup, registering at 3.21 ± 0.28% as opposed to 0.97 ± 0.12% for controls (*p* < 0.001). Yet, the pre-incubation with an anti-PD-1 antibody attenuated the stimulatory effect of AdMSC-PDL1-Akt on regulatory T cells, with a resultant expression of 1.96 ± 0.16% (*p* = 0.018, relative to AdMSC-PDL1-Akt co-cultivated T cells).

## 3. Discussion

### 3.1. Principal Findings

The current investigation presents a compelling set of findings that underscore the potential of AdMSCs, strategically optimized through genetic engineering to overexpress PD-L1 and Akt, in fostering improved cardiac functionality following an AMI. The engineered cells exhibit significant mitigation effects on the detrimental process of ventricular remodeling post-AMI, which is a vital determinant in the overall prognosis and recovery trajectory of affected patients. The unveiled superiority of AdMSC-PDL1-Akt over traditional therapies is most evident in their robust resistance to oxidative stress and their heightened proliferative capacity. These characteristics collectively improve the likelihood of effective cellular survival and engraftment within the harsh environment of the infarcted myocardium, laying a promising foundation for future cardiac dysfunction therapeutic strategies.

Indeed, we initiated a proof-of-concept pilot study to assess the AKT-PD-L1 dual gene engineering approach. The overexpression of PD-L1 and Akt in adipose-derived mesenchymal stem cells (AdMSCs) is of considerable interest in the context of myocardial infarction (MI) due to their respective potential roles in modulating immune responses and enhancing cell survival. PD-L1 may contribute to immune regulation, whereas the activation of Akt is associated with pro-survival and anti-apoptotic effects. These functions could be particularly relevant in MI, where both cell survival and immune responses are crucial.

The potential rationale behind overexpressing PD-L1 in AdMSCs and the possible mechanism of action include (1) Immunomodulation and immune evasion: PD-L1 is known for its role in immune modulation. It interacts with PD-1 on immune cells, sending inhibitory signals and suppressing immune responses. Overexpressing PD-L1 in AdMSCs could enhance the immunomodulatory properties of these cells. This might be especially relevant in conditions where immune responses need to be finely regulated, such as inflammatory states post-infarct. (2) Anti-inflammatory effects: PD-L1 has been associated with anti-inflammatory effects, and its upregulation might contribute to reducing inflammation. In a context like post-infarct, where inflammation plays a role, AdMSCs overexpressing PD-L1 could potentially have a more significant anti-inflammatory impact. (3) Synergistic effects with Akt activation: considering the synergy between PD-L1 and Akt, overexpressing PD-L1 in AdMSCs alongside Akt activation may amplify the immunomodulatory and anti-inflammatory effects. Akt activation may enhance cell survival and proliferation, while PD-L1 could contribute to the suppression of excessive immune responses. (4) Therapeutic potential in inflammatory conditions: overexpression of PD-L1 in AdMSCs might be explored as a therapeutic strategy in inflammatory diseases, where precise control over immune responses is important. This approach could potentially regulate the balance between promoting cell survival and modulating immune responses, contributing to tissue repair and homeostasis. (5) Enhanced therapeutic efficacy in cell-based therapies: AdMSCs are increasingly investigated for their therapeutic potential in regenerative medicine. Overexpressing PD-L1 may enhance the efficacy of AdMSC-based therapies by optimizing their immunomodulatory properties. This could be particularly beneficial in scenarios where transplanted cells face potential immune rejection. In summary, overexpressing PD-L1 in AdMSCs could be a strategic approach to harness the immunomodulatory and anti-inflammatory properties of both PD-L1 and Akt signaling pathways. This may have potential applications in conditions where balancing immune responses and promoting tissue repair are critical, such as in post-infarct scenarios or other inflammatory conditions. Further experimental studies would be necessary to validate the potential mechanism of action.

### 3.2. Strengths of PD-L1 and AKT Dual Overexpressing AdMSC Therapy in AMI Model

The coinhibitory pathway involving PD-1 and PD-L1 serves a critical function in the establishment of immune tolerance by mitigating unwarranted immune cell activation. Therapeutic agents targeting the PD-1/PD-L1 axis have emerged as promising anticancer interventions [32]. Both preclinical and clinical evidence robustly support the notion that PD-1/PD-L1 coinhibitory signaling can temper excessive inflammation and confer protective effects on coronary vessels against atherosclerotic damage [33].

Conversely, the expression of PD-L1 is governed by a complex interplay of factors, including various signaling pathways, transcriptional regulators, and epigenetic mechanisms [34]. The Phosphoinositide 3-kinase/Protein kinase B (PI3K/Akt) pathway has been identified as one of the key signaling routes capable of modulating PD-L1 expression through either transcriptional or post-transcriptional means [35]. Inhibition of Akt has been shown to result in diminished PD-L1 expression, substantiating the role of Akt as a positive regulator of PD-L1 [36]. The Akt signaling pathway has been implicated in the regulation of various cellular processes, including cell survival, proliferation, and immune responses. There is evidence supporting the role of Akt as a positive regulator of PD-L1 expression. The relationship between Akt and PD-L1 expression, particularly in the context of various pathological conditions, including cancer and inflammatory diseases, has been investigated. Studies have demonstrated that the Akt signaling pathway can positively regulate the expression of PD-L1. Inhibition of Akt has been shown to result in decreased levels of PD-L1 expression in cancer cells. On the contrary, Akt activation may lead to an increase in PD-L1 expression. Akt-mediated upregulation of PD-L1 suggests a potential link between cell survival signaling and immune modulation. This connection is particularly relevant in the context of diseases like inflammatory conditions, such as post-infarct. The overexpression of PD-L1 and Akt in AdMSCs could exert a synergistic effect on the modulation of immune and inflammatory response [37]. Additionally, Akt serves as a pivotal molecule in the anti-apoptotic pathways within cardiomyocytes [38]. Activation of Akt has been demonstrated to preserve cardiac function and mitigate damage following transient cardiac ischemia in vivo [26]. Moreover, the initiation of reperfusion injury salvage kinase (RISK) signaling via Akt, triggered by β2-adrenergic receptor stimulation, has been shown to result in a reduction in infarct size in rodent models [39].

The initial phase of our research aimed to authenticate the overexpression of Akt and PD-L1 in AdMSC post-transduction, leveraging both immunofluorescence and Western blot analyses as reliable verification methods. The pronounced presence of PD-L1 and Akt in the transduced cells (pUltra-PD-L1-Akt AdMSCs), as evidenced through these rigorous tests, validates the success and stability of the transduction process in upregulating these proteins. Notably, this enhanced expression was associated with significant functional improvements in the AdMSCs, including a remarkable increase in the cell proliferation rate (92.80 ± 3.64%) and heightened resistance to oxidative stress induced by hydrogen peroxide exposure. These enhancements in cellular functions are crucial, given the hostile and oxidative stress-rich environment of infarcted myocardial tissue, where these cells are expected to operate and exert their therapeutic effects.

Upon establishing the dual overexpression of Akt and PD-L1 in AdMSCs and the enhancements in cellular functions in vitro, we proceeded to evaluate their therapeutic potential in an in vivo rat model of MI. The application of these cells showed a promising reduction in infarct size and mitigated the myocardial damage induced by MI. This study observed a notable improvement in various cardiac function parameters, such as the ESPVR and PRSW, suggesting improved systolic function, which is indicative of the therapeutic potential of the AdMSC-PD-L1-Akt cells in a post-MI scenario.

### 3.3. Mechanistic Insights from PD-L1 and AKT Dual Overexpressing Cell Therapy

The inflammatory and apoptotic cascades activated post-MI play a pivotal role in determining the extent of myocardial damage and the prognosis of the condition [12,40]. In our study, the administration of AdMSC-PD-L1-Akt cells was associated with a significant downregulation of caspase-3 and NFκB, markers indicative of apoptosis and inflammation, respectively. This downregulation not only highlights the anti-apoptotic and anti-inflammatory properties of the transduced cells but also underscores their potential to limit the extent of myocardial damage post-MI.

The immunomodulatory role of the AdMSC-PD-L1-Akt cells was further evidenced by the upregulation of CD25 and the significant induction of CD4^+^CD25^+^ regulatory T cells. Regulatory T cells (Tregs) play a vital role in immune modulation and tissue repair and have been implicated in the improvement in outcomes in various inflammatory and autoimmune conditions [41,42,43]. The observed increase in CD25 expression and the induction of Tregs suggest that the therapeutic effects of AdMSC-PD-L1-Akt cells may, in part, be mediated through the modulation of the immune response post-MI, creating a more conducive environment for myocardial repair and regeneration.

The rationale for investigating PD-L1 and Akt in the context of MI involves their roles in immune regulation and cell survival, respectively. PD-L1 is part of the coinhibitory pathway involving PD-1, and this pathway plays a vital role in immune tolerance. In the context of MI, there is an inflammatory response that involves the activation of immune cells. Excessive or prolonged immune activation can contribute to tissue damage. PD-L1, when binding to its receptor PD-1 on immune cells, inhibits their activation, preventing unwarranted and potentially harmful immune responses. Modulating this pathway could be a strategy to adjust the immune response in MI and potentially attenuate tissue damage. Akt plays a key role in cell survival and anti-apoptotic signaling. In the context of MI, there is a significant risk of cell death (apoptosis) due to ischemic injuries. Activating the Akt pathway can support cell survival and inhibit apoptosis, potentially protecting cardiac cells from damage during and after an infarction [44].

Akt can influence the expression and function of CD25, which is the alpha subunit of the interleukin-2 receptor (IL-2R) and is often used as a marker for Tregs. Akt activation can promote the production of IL-2, a key cytokine involved in T cell proliferation and survival. CD25 is the alpha chain of the IL-2 receptor, and its expression is critical for high-affinity binding of IL-2. Akt activation may enhance IL-2 signaling, leading to increased CD25 expression on T cells. PD-L1 is a transmembrane protein commonly expressed on the surface of antigen-presenting cells and tumor cells but not on AdMSCs. PD-L1 has been recognized as playing a role in immune regulation, and its interaction with PD-1 on T cells is a key aspect of this regulation. While PD-L1 itself does not directly upregulate CD25^+^ T cells, its engagement with PD-1 contributes to the modulation of T cell activation and function, including Tregs. MI triggers an inflammatory response involving the infiltration of immune cells. Excessive inflammation can contribute to tissue damage. PD-L1, when expressed on the surface of AdMSCs, may interact with PD-1 on immune cells, inhibiting their activation. This interaction can potentially modulate the immune response, preventing an exaggerated and harmful inflammatory reaction in the infarcted area [45,46].

Here are some key ways in which Tregs contribute to immune regulation post-MI: (1) Tregs suppress the activation and function of effector T cells, preventing an exaggerated and potentially harmful inflammatory response in the infarcted tissue. (2) Tregs influence macrophage polarization toward an anti-inflammatory phenotype, contributing to tissue repair and resolution of inflammation. (3) Tregs secrete anti-inflammatory cytokines such as IL-10 and TGF-β, which are involved in tissue repair and remodeling. (4) Tregs control the recruitment and activation of other immune cells, influencing the overall composition of the immune response in the infarcted heart. (5) Tregs promote angiogenesis, the formation of new blood vessels, which is crucial for the restoration of blood supply to the infarcted area [47]. In the current study, the observed increase in CD25 expression and the induction of Tregs suggest that the therapeutic effects of AdMSC-PD-L1-Akt cells may, in part, be mediated through the modulation of the immune response post-MI, creating a more conducive environment for myocardial repair and regeneration. Understanding the regulatory functions of Tregs in the aftermath of MI is critical for developing therapeutic strategies that harness the immunomodulatory potential of these cells to enhance cardiac repair and mitigate adverse remodeling.

### 3.4. Limitations of the Study

While the present study provides valuable insights into the efficacy of genetically optimized AdMSCs overexpressing PD-L1 and Akt in improving post-MI cardiac function, several limitations warrant careful consideration. First, our study predominantly focuses on short-term post-MI outcomes; hence, the long-term efficacy and stability of AdMSC-PDL1-Akt therapy remain unexplored. Understanding the sustainability of the observed therapeutic benefits over extended periods is crucial for validating this approach as a viable long-term treatment strategy for post-MI cardiac dysfunction. Second, while the overexpression of PD-L1 and Akt has shown promising results, the specific molecular mechanisms and interactions driving these positive outcomes need to be elucidated further. A more in-depth exploration into the mechanistic underpinnings would offer a clearer understanding of the therapy’s mode of action and potential side effects. Third, the study did not investigate the potential impact of varying degrees of PD-L1 and Akt overexpression on therapeutic outcomes, which may provide additional valuable insights for optimizing the genetic engineering process. Fourth, we did not quantify the number of infiltrated T cells in our experiment. Consequently, we relied on the total cell count in the slides for justification. Without blood or tissue slide samples remaining, we could not check the total number of infiltration T cells, but according to our recent study, PDL1 and AKT-modified umbilical cord mesenchymal stem cells (UMSC-PD-L1-AKT) could also enhance T-regulatory cells in the ischemic brain without increasing T cell infiltration [48]. Thus, we believe that our result might be convincing. Lastly, we acknowledge that the sample size in our experimental groups is relatively small, which may limit the statistical power and generalizability of the study findings. Future studies with larger sample sizes and diverse models are required to corroborate and build upon the findings reported here. These studies should also explore the potential interactions between AdMSC-PDL1-Akt therapy and standard post-MI pharmacological interventions to evaluate the feasibility and safety of combined treatment approaches. Indeed, the path from the current encouraging pre-clinical findings to the development of effective clinical therapies is complex and requires careful navigation through further studies and clinical trials.

## 4. Methods and Materials

### 4.1. Preparation, Isolation, and Characterization of AdMSCs

Human adipose tissue samples, obtained with approval from the Institutional Review Board (IRB) of China Medical University Hospital (CMUH109-REC1-153 (CR-3)), were subjected to triple washings using phosphate-buffered saline (PBS) devoid of calcium and magnesium ions (Ca^2+^ and Mg^2+^-free PBS; Life Technology, Carlsbad, CA, USA). The tissues were subsequently mechanically dissected into fragments with dimensions less than 0.5 cm^3^. These tissue fragments were treated with collagenase type 1 (Sigma-Aldrich, St. Louis, MO, USA) and incubated for a duration of three hours at 37 °C within a humidified atmosphere consisting of 95% air and 5% CO_2_. Explants were cultured in NutriStem^®^ MSC XF Medium supplemented with NutriStem^®^ XF Supplement Mix (Sartorius, Goettingen, Germany) and 5% UltraGRO™-Advanced-PURE Cell Culture Supplement (AventaCell Biomedical, Kent, WA, USA), along with antibiotics, in a 5% CO_2_ environment at 37 °C. The explants were left undisturbed for 5–7 days to facilitate cell migration. Following 4–8 passages, the adipose tissue-derived mesenchymal stem cells (ADMSCs) exhibited a homogenous spindle-shaped morphology. We characterized the specific surface molecules of these cells using flow cytometry. Cells, detached using TrypLE™ Select Enzyme (Gibco, Waltham, MA, USA) and washed with PBS, were incubated with antibodies (including CD73, CD90, CD105, CD19, CD34, CD45, CD11b, and HLA-DR) conjugated with fluorescein isothiocyanate (FITC) or phycoerythrin (PE) (BD Biosciences, Franklin Lakes, NJ, USA). Analysis was performed using a Becton Dickinson flow cytometer and FlowJo v.7.6 software.

### 4.2. Lentiviral Vector Construction and Transfection Protocols

Lentiviral vectors (pLAS3w), along with packaging (psPAX2) and envelope (pMD2.G) plasmids, were acquired from Academia Sinica, Taiwan. Complementary DNA (cDNA) sequences encoding the full-length human Akt, PD-L1, Luciferase (Luc), and a control green fluorescent protein (GFP) were sourced from existing cDNA constructs (pCMV6-myc-DDK-Akt, pCMV6-myc-DDK-PD-L1, pRMT-Luc, and OriGene, respectively). These sequences were sub-cloned into the pUltra vector (Addgene, Watertown, MA, USA) using specific restriction enzyme linkers (EcoR1 and Nhe1 for Akt; BamH1 and Not1 for PD-L1; EcoR1 and Nhe1 for Luc) to generate constructs such as pUltra-Akt-PD-L1, pUltra-Akt-GFP, and pUltra-PD-L1-GFP.

The constructed templates were then amplified via polymerase chain reaction (PCR) using sequence-specific primers, followed by digestion with the corresponding restriction enzymes. Subsequently, the digested fragments were sub-cloned into the lentiviral backbone vector plasmid pLAS3w (Academia Sinica, Taipei, Taiwan). To generate the recombinant lentiviruses carrying Akt-PD-L1, Akt, PD-L1, Luc, and control GFP, co-transfection was performed using the recombinant plasmid and packaging and envelope plasmids in 293T cells at a ratio of 3:3:1. Transfection was executed using XtremeGene HP DNA transfection reagent (Roche, Basel, Switzerland).

AdMSCs were plated at a density of 1 × 10^5^ cells per well in a 6-well plate, each well containing a final volume of 1 mL. The cells were transduced at a multiplicity of infection (MOI) of 5, in triplicate. Protamine sulfate, procured from Sigma-Aldrich and prepared as a 5 mg/mL stock solution in DMEM-LG, was added to achieve the targeted final concentration. Following a 24 h incubation period for transduction, the medium was replenished with 1.5 mL per well to establish cell populations termed AdMSC-PDL1-Akt, AdMSC-Akt-GFP, AdMSC-Luc, and AdMSC-PD-L1-GFP. For the selection of successfully transduced cells, the overconfluent monolayers were sub-cultured into fresh 6-well plates and subjected to selection pressure through the addition of either 1.0 mg/mL G418 or puromycin solutions (Sigma-Aldrich, St. Louis, MO, USA).

### 4.3. Characterization of AdMSC-PDL1-Akt Cells: Immunophenotyping, Proliferation, and Viability Assessment

The efficacious co-transfection of PD-L1 and Akt in AdMSCs was substantiated through immunofluorescence staining and Western blot analyses, employing specific antibodies sourced from Cell Signaling Technologies (Danvers, MA, USA). Nuclear counterstaining was achieved with 4′,6-diamidino-2-phenylindole (DAPI; DAPI Fluoromount-G, SouthernBiotech, Birmingham, AL, USA).

Cellular proliferation rates for both the native AdMSCs and the AdMSC-PDL1-Akt cells were ascertained using Carboxyfluorescein Succinimidyl Ester (CFSE; CellTrace™, ThermoFisher Scientific, Waltham, MA, USA). Cells were incubated in 5 mM CFSE at 37 °C for 10 min, subsequently quenched with fetal bovine serum (FBS), and subjected to flow cytometric analysis for quantification.

To evaluate cellular viability, we employed the WST-8 assay [2-(2-methoxy-4-nitrophenyl)-3-(4-nitrophenyl)-5-(2,4-disulfophenyl)-2H-tetrazolium, monosodium salt; Cell Counting Kit-8, MedChem Express, Monmouth Junction, NJ, USA]. Both ADMSC and ADMSC-PDL1-Akt cells were plated in a 96-well format and exposed to 1, 3, and 10 μM concentrations of H_2_O_2_ to induce cellular stress. Absorbance at 450 nm was recorded after 24 h using a Tecan microplate reader (Tecan, Männedorf, Switzerland). Cell viability ratios were computed as follows: Cell Viability (%) = (Optical Density of Treated Cells/Optical Density of Control Cells) × 100.

Intracellular levels of reactive oxygen species (ROS) were assessed utilizing the cell-permeant 2′,7′-dichlorodihydrofluorescein diacetate (H2DCFDA) assay, conducted in accordance with the manufacturer’s guidelines (MedChem Express, Monmouth Junction, NJ, USA). ROS generation was quantified through absorbance measurements at 525 nm, taken 20 to 120 min post-exposure to 10 μM H_2_O_2_, utilizing a Tecan microplate reader (Tecan, Männedorf, Switzerland).

### 4.4. Animal Experimental Design and Protocols

All animal experiments received approval from the Institutional Animal Care and Use Committee at China Medical University (Approval No. 2019-362-1) and were executed in strict accordance with the guidelines stipulated by the U.S. National Institutes of Health’s Guide for the Care and Use of Laboratory Animals. Male Wistar rats, 8 weeks old, were procured from BioLASCO Taiwan Co., Ltd. (Taipei, Taiwan) and were randomly allocated into four distinct experimental groups: sham-operated (Sham), myocardial infarction (MI), MI followed by AdMSC injection (MI+AdMSC), and MI followed by AdMSC-PDL1-Akt injection (MI+AdMSC-PDL1-Akt), with *n* = 7 for each group (Figure 7).

Animals were anesthetized using 2% isoflurane (Abbott, Abbott Park, IL, USA) and mechanically ventilated using a specialized rodent ventilator (New England Medical Instruments, Medway, MA, USA). Myocardial infarction was induced via the permanent ligation of the left anterior descending coronary artery (LAD) employing a silk suture. Sham-operated animals were subjected to identical surgical protocols, excluding the LAD ligation. Verification of successful LAD ligation was achieved through observing ST-segment elevations in lead II electrocardiograms, which were acquired using a PowerLab data acquisition system (ADInstruments, Colorado Springs, CO, USA) coupled with Chart v7.3.8 software. Upon confirmation of MI induction, AdMSCs at a dose of 1 × 10^6^ in 1 mL phosphate-buffered saline were administered through intramyocardial injections targeted around the peri-infarct zone.

### 4.5. Pressure–Volume Loop Hemodynamic Analysis

For hemodynamic evaluations, rats were subjected to anesthesia via intraperitoneal administration of thiopental sodium at dosages ranging from 60 to 80 mg/kg. Subsequent to anesthesia, a surgical incision was made to expose the right carotid artery, facilitating the insertion of a 2.0 F microtip pressure–volume (PV) catheter (Model SPR-838, Millar Instruments, Houston, TX, USA). Following initial arterial pressure recording, the catheter was carefully advanced into the left ventricle (LV) under guidance from real-time pressure waveforms. Upon stabilization, continuous pressure and volume signals were recorded through a specialized PV conductance system (MPVS Ultra, emka TECHNOLOGIES, Paris, France) interfaced with a digital converter unit (Model ML-870, ADInstruments, Colorado Springs, CO, USA). Hemodynamic variables were measured across varying preloads induced by transient mechanical compression of the abdominal inferior vena cava. To ascertain preload-independent parameters, transient compressions of the abdominal inferior vena cava were executed concomitantly with PV loop data acquisition.

### 4.6. Infarct Size Quantification

Upon completion of experimental protocols, subjects were anesthetized utilizing 5% isoflurane and subsequently euthanized via cervical dislocation. Excised hearts were promptly mounted on a Langendorff apparatus and perfused with isotonic saline, followed by the administration of phthalocyanine blue dye to delineate non-ischemic tissue. The cardiac tissue was sectioned into 1 mm slices, which were then incubated in 1% 2,3,5-triphenyltetrazolium chloride (TTC) solution at 37 °C for a period of 30 min. This procedure allowed for the differentiation between infarcted (depicted as white areas) and non-infarcted (depicted as red areas) zones within the area at risk (AAR), identified as regions negative for phthalocyanine blue staining. Infarct size was quantified by digital analysis of slice images, employing ImageJ software (Version 1.51j8) for this purpose.

### 4.7. Immunofluorescence Analysis for Cardiac Tissue Characterization

Serial cross-sections of myocardial tissues were subjected to immunofluorescence staining employing antibodies against Caspase-3 (PROTEINTECH, Chicago, IL, USA), CD25 (Abcam, Cambridge, UK), NF-κB (Abcam, Cambridge, UK), and myosin heavy chain (Leica, Wetzlar, Germany). Subsequently, tissue sections were incubated with the appropriate fluorophore-conjugated secondary antibodies to assess in situ protein expression levels. For semi-quantitative analysis of Caspase-3, CD25, NF-κB, and myosin heavy chain expression, image data were collected from four independent samples per experimental group, with two images scrutinized per sample. Staining intensity was evaluated on a scale of 0 to 4, where a score of 0 represented an absence of staining, and a score of 4 indicated maximal staining intensity.

### 4.8. Isolation of Murine CD4^+^ T Lymphocytes and Co-Culture with Adipose-Derived Mesenchymal Stem Cells

Adult C57BL/6 mice were acquired from the National Laboratory Animal Center, Taiwan. CD4^+^ T cells were isolated from splenic homogenates of these mice utilizing a negative-selection protocol with the BD iMag Mouse CD4 T Lymphocytes Enrichment Set (BD Biosciences, Franklin Lakes, NJ, USA). Post-isolation purity of CD4^+^ T cells exceeded 93%, as verified via flow cytometry. Following isolation, the CD4^+^ T cells concentration was adjusted to 5 × 10^5^ cells/mL/well. These isolated lymphocytes were subsequently co-cultured with either unmodified AdMSC or AdMSC engineered to overexpress PD-L1 and Akt (AdMSC-PDL1-Akt), at a density of 4 × 10^4^ cells/well. The co-culture was maintained in 12-well transwell tissue culture plates with culture medium for AdMSCs described previously for a period of five days. Proportions of CD4^+^CD25^+^ T lymphocytes were quantitatively assessed both prior to and following the co-culture period with AdMSC and AdMSC-PDL1-Akt.

### 4.9. Flow Cytometric Analysis of CD4^+^CD25^+^ Regulatory T Cells

To assess the potential modulatory effects of stem cells on the expansion of regulatory T cells, the frequencies of CD4^+^CD25^+^ T cells were quantified using flow cytometry at both baseline (day 0) and at day 4 post-co-culture. Cells were stained for surface markers CD4^+^ and CD25^+^ utilizing fluorochrome-conjugated monoclonal antibodies specific for mouse CD4^+^ (allophycocyanin-conjugated) and mouse CD25^+^ (BB515A-conjugated), both sourced from BD Biosciences. Flow cytometric analyses were conducted using a FACS Canto flow cytometer (BD Biosciences), and the resulting data were processed using FlowJo analytical software (Version 7.6, Tree Star Inc., Ashland, OR, USA). Additionally, an anti-mouse PD-1 antibody was employed as an inhibitory agent to disrupt the PD-1/PD-L1 interaction, sourced from Bio X Cell (Lebanon, NH, USA).

### 4.10. Statistical Analysis

All data were acquired in a double-blinded manner to mitigate bias. Quantitative results are presented as mean ± standard error of the mean (SEM). To compare means between two independent groups, two-tailed Student’s *t*-tests were employed. For comparisons involving multiple groups, a two-way analysis of variance (ANOVA) followed by a Newman–Keuls post hoc analysis was conducted. Intensity scoring for immunofluorescence analysis of cardiac tissues was conducted by two independent researchers who were blinded to the experimental conditions. Compiled intensity scores were averaged and subjected to statistical analysis using the Kruskal–Wallis test. Statistical significance was established at a *p*-value of less than 0.05.

## 5. Conclusions

The present study provides pivotal insights into the therapeutic efficacy of AdMSCs overexpressing PD-L1 and Akt in a rat model of MI. The results underscore a significant improvement in post-MI cardiac function attributed to the enhanced resilience, proliferation, and immunomodulatory effects of the engineered stem cells. Specifically, the upregulation of CD25^+^ T cells plays a cardinal role in this therapeutic landscape, opening new horizons for innovative and effective treatment modalities for patients grappling with post-MI cardiac dysfunction.

## Figures and Tables

**Figure 1 ijms-25-00134-f001:**
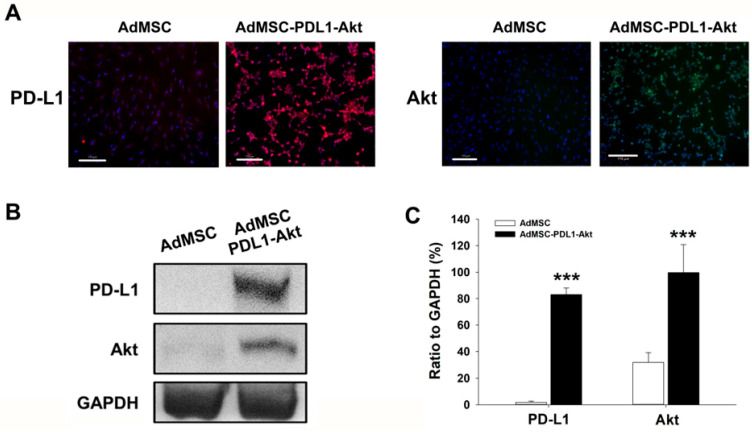
Enhanced Expression of PD-L1 and Akt in AdMSCs. Immunofluorescence was employed to assess the expression levels of PD-L1 and Akt in both AdMSC and AdMSC-PDL1-Akt cells using specific probes. (**A**) Cell nuclei were visualized by counterstaining with DAPI, depicted in blue. Additionally, the protein abundance of both PD-L1 and Akt was ascertained in cell lysates via Western blot analysis (**B**) and subsequently quantified. (**C**) GAPDH served as an internal loading control. Scale bar: 170 μm. All data are represented as mean ± SEM. *** *p* < 0.01 when compared to AdMSC.

**Figure 2 ijms-25-00134-f002:**
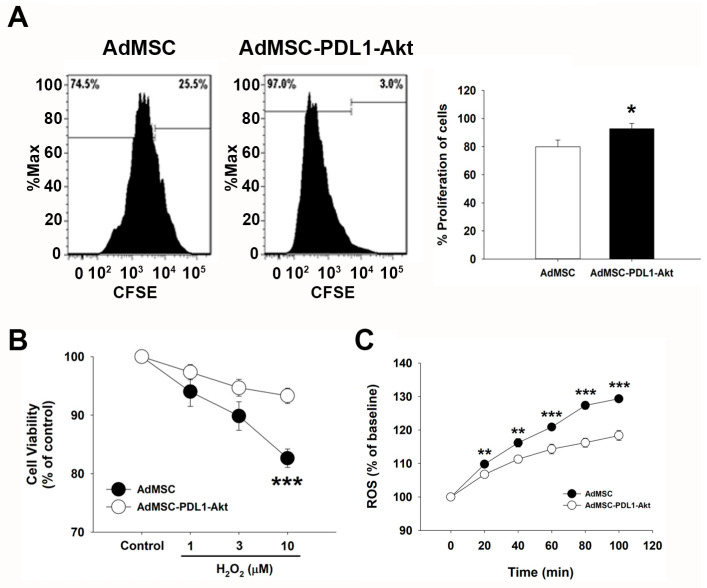
Enhanced proliferation and anti-oxidative capability of AdMSC-PDL1-Akt. (**A**) Proliferative capacities of AdMSC and AdMSC-PDL1-Akt were assessed using the CFSE assay, with results visualized by flow cytometry. (**B**) Cellular viability was evaluated through the WST-8 assay. Concentrations of 1, 3, and 10 μM H_2_O_2_ were introduced to AdMSC (black circle) and AdMSC-PDL1-Akt (white circle) to provoke cellular death. (**C**) ROS production, induced by 10 μM H_2_O_2_, was juxtaposed between AdMSC (black circle) and AdMSC-PDL1-Akt (white circle) using the H2DCFDA method at various time intervals. All data are represented as mean ± SEM. * *p* < 0.05; ** *p* < 0.02; *** *p* < 0.01 when compared to AdMSC.

**Figure 3 ijms-25-00134-f003:**
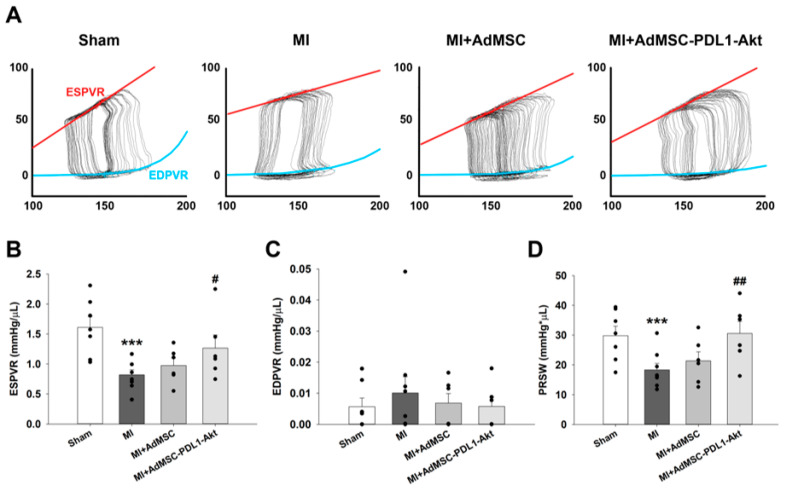
AdMSC-PDL1-Akt ameliorates MI-induced systolic dysfunction in rat hearts following left anterior descending artery (LAD) ligation. (**A**) Representative pressure–volume (P–V) loops across varied preloads for sham, MI, MI+AdMSC, and MI+AdMSC-PDL1-Akt rat models. (**B**) Comparative analysis of the average slopes of the end-systolic pressure–volume relationship (ESPVR). (**C**) Comparison of the average slopes of the end-diastolic pressure–volume relationship (EDPVR) across the four rat models. (**D**) The preload recruitable stroke work (PRSW) is also juxtaposed among the four groups. All data are presented as mean ± SEM. For quantitative analysis, the sample size was *n* = 6 for each group, and each black dot indicates individual sample. *** *p* < 0.01 in comparison to sham rats; ^#^ *p* < 0.05, ^##^ *p* < 0.02 when compared to MI rats.

**Figure 4 ijms-25-00134-f004:**
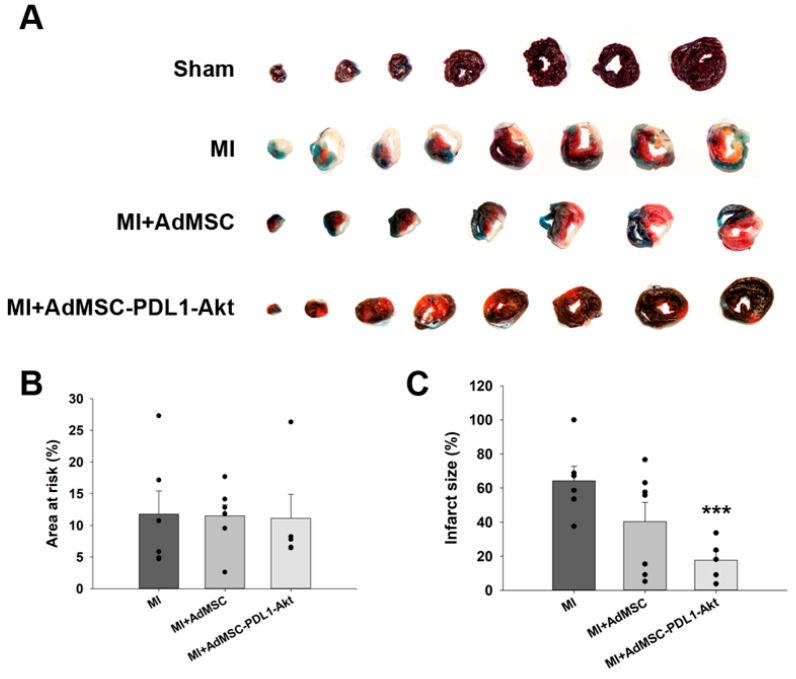
AdMSC-PDL1-Akt reduces infarct size progression in rat hearts following LAD ligation. Following LAD ligation, rat hearts were perfused with phthalocyanine blue dye and subsequently immersed in triphenyltetrazolium chloride (TTC). (**A**) Displayed are representative heart section images from sham, MI, MI+AdMSC, and MI+AdMSC-PDL1-Akt rats. (**B**) Quantitative comparison of the area at risk (designated by the nonblue region). (**C**) Quantitative assessment of the infarct size (indicated by the white region). All data are articulated as mean ± SEM. Quantitative analysis was based on a sample size of *n =* 6 for each group, and each black dot indicates individual sample. *** *p* < 0.01 in comparison to MI rats.

**Figure 5 ijms-25-00134-f005:**
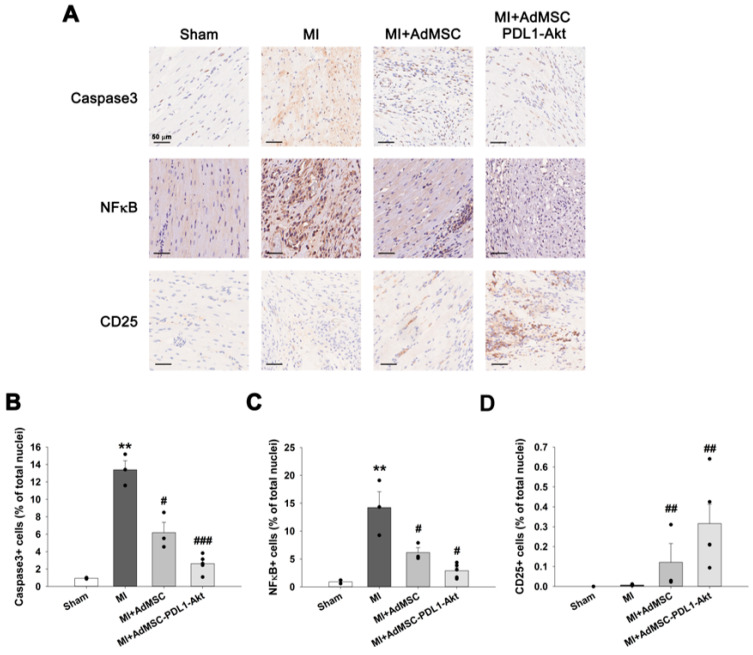
AdMSC-PDL1-Akt downregulates caspase3 and NFκB expression and upregulates CD25^+^ T cell expression. (**A**) Immunohistochemical analysis illustrating the prevalence of caspase3, NFκB, and CD25^+^ T cells in myocardial cross-sections from sham, MI, MI+AdMSC, and MI+AdMSC-PDL1-Akt rats. (**B**) Quantitative assessment of the proportion of cells demonstrating positive staining for caspase3. (**C**) Quantitative assessment of the proportion of cells demonstrating positive staining for NFκB. (**D**) Quantification of cells exhibiting positive CD25 expression. All data are presented as mean ± SEM. ** *p* < 0.02 when compared to sham; ^#^ *p* < 0.05, ^##^ *p* < 0.02, ^###^ *p* < 0.01 in comparison to MI rats.

**Figure 6 ijms-25-00134-f006:**
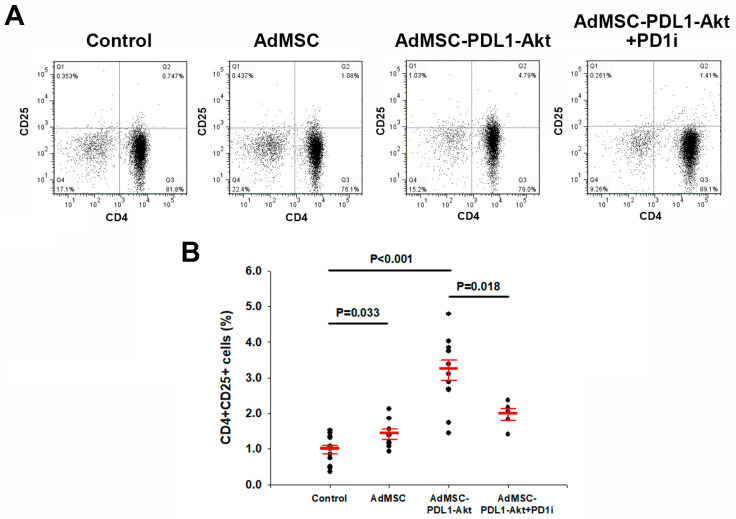
Augmented CD4^+^ CD25^+^ regulatory T cell proportions following co-culture with AdMSC-PDL1-Akt. (**A**) Depiction of CD4^+^CD8^+^ T cell proportions (region Q2) from murine spleen, representing findings from ten distinct experiments. Comparative assessment of T cells co-cultured with either AdMSC or AdMSC-PDL1-Akt over a span of 4 days against a baseline control. Utilization of PD1 inhibitor (Anti-mouse PD-1 antibody) demonstrated blockade of AdMSC-PDL1-Akt effects. (**B**) Observed enrichment of regulatory T cell proportions (CD4^+^CD25^+^ T cells) post 4-day co-culture with AdMSC-PDL1-Akt, in contrast to AdMSC co-culture. All data are denoted as mean ± SEM.

**Figure 7 ijms-25-00134-f007:**
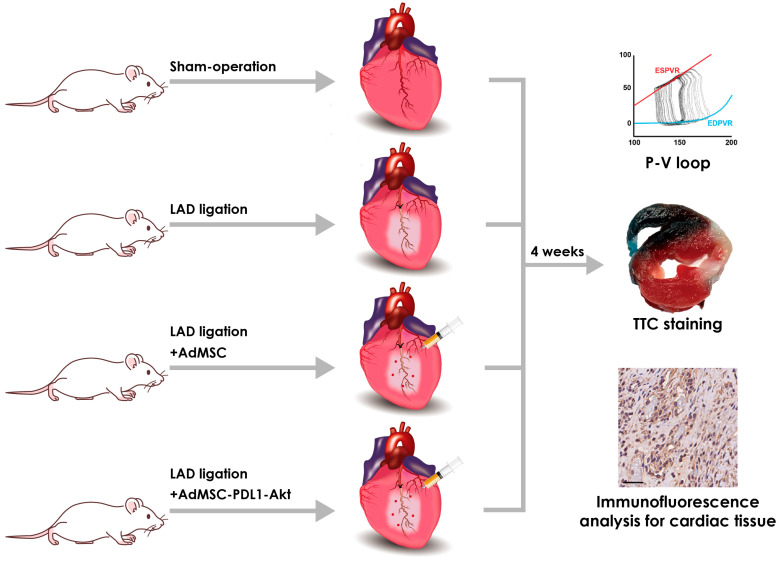
Animal experimental design and protocols. Male Wistar rats were randomly allocated into four distinct experimental groups: sham-operated (Sham), myocardial infarction (MI), MI followed by AdMSC injection (MI+AdMSC), and MI followed by AdMSC-PDL1-Akt injection (MI+AdMSC-PDL1-Akt). MI was induced via the permanent ligation of the left anterior descending coronary artery (LAD) employing a silk suture. Scale bar: 50 μm.

## Data Availability

The data that support the findings of this study are available from the corresponding author upon reasonable request.

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
