# Peer review of "PD-L1 and AKT Overexpressing Adipose-Derived Mesenchymal Stem Cells Enhance Myocardial Protection by Upregulating CD25+ T Cells in Acute Myocardial Infarction Rat Model"

_ijms, 2023, doi:10.3390/ijms25010134_

Round 1

Reviewer 1 Report

Comments and Suggestions for Authors

In the present study, the authors have used PD-L1Akt overexpressed ADMSCs and check its effect after MI by injecting these cell lines in Rats having MI surgery.

1.     Please give some background in one or two line why you have chosen Programmed Death Ligand 1 (PD-L1) and Protein Kinase B (Akt) overexpression in adipose-derived mesenchymal stem cells (AdMSCs) in context to MI

2.     The authors have just mentioned “The coinhibitory pathway involving Programmed Death Receptor 1 (PD-1) and Programmed Death Ligand 1 (PD-L1) serves a critical function in the establishment of immune tolerance by mitigating unwarranted immune cell activation” Please elaborate the rationale behind why PDL1 and Akt in MI. Also please provide proper references behind the selection of these targets.

3.     Can the author use in vitro model of cardiovascular pathologies like MI: Macrophage treatment with OxLDL and native LDL is an accepted model.

4.     The authors have conducted the experiments using overexpression approach. Could you also validate your finding by silencing approach. If not in vivo, then at least in vitro.

5.     Please indicate why you have used CD25 alongside NFKB and Caspase3 in results for the readers to properly understand.

6.     Also explain in results that how inflammatory pathway is being ameliorated in the overexpressed group.

7.     Have the authors checked the proliferation of T cells in their experimental model? If not, then they should considering checking the proliferation to explain how C25 positive regulatory T cells are getting upregulated after MI

8.     Please also check some of the inflammatory genes apart from NFKB.

9.     What are the targets of NFKB in your experimental settings.

Reviewer 2 Report

Comments and Suggestions for Authors

Comments:

1)      Method: Section ‘2.1. Preparation, isolation and characterization of AdMSCs’, looks incomplete and misleading. Section mentioned characterization of AdMSCs, but how cells were cultured, media growth factors and how it was characterized using any specific markers, all these information are lacking.

2)      Considering comment#1, it is highly recommended to provide the information related to AdMSCs culture and its characterization.

3)       Figure 1, It will be interesting to fortify the Figure 1 with phosphorylated form of AKT (p-AKT)?

4)      Section 2.4. Animal Experimental Design and Protocols, not explained properly. Provide a figure and explain the time points, when MI induced, when cells were administered, when heart harvested?

5)      In Fig. 5, CD25 expression in MI+AdMSC-PDL1-Akt heart section showed more cell infiltration compared to MI+AdMSC? How authors can justify this?

6)      How T cells and AdMSC co-cultured was maintained?Total cell harvested from culture?  or only T cells?

7)      Figure 6, Provide data of PDi alone? i.e. (Control + PDi). Ideally authors should have provided effect of PDi  with AdMSC also.

8)      Rationale of overexpressing PD-L1 in AdMSC is not clearly defined. It should be thoroughly discussed to explain the mechanism of action.

9)      Discussion: line 25-26, reference 36 is poorly explained, which cells, in what context? In immune cells or  AdMSC?

10)  There are various limitations of the study. No data to support the PD-L1 and AKT mediated mechanism of action.PD-L1 overexpression induced the surface expression of PD-L1? How PD-L1 expression on AdMSC interacted with T cells, No expression of PD-1 expression checked on effector T cells? Cite appropriate reference or data to explain the probable mechanism. Data is too inconclusive to explain the precise mechanism and discussion has also not addressed the same.

Comments on the Quality of English Language

Minor English editing required.

Reviewer 3 Report

Comments and Suggestions for Authors

This study by Yu-Kai Lin and Lien-Cheng Hsiao presented data on protective effects of PD-L1 and AKT overexpressing adipose-derived mesenchymal stem cells (AdMSC PD-L1 AKT) in myocardial infarction. By emphasizing in vitro and in vivo models, the authors found that the PD-L1 AKT approach showed reduced ROS production and improved systolic dysfunction, followed by MI. Finally, authors showed that AdMSC PD-L1 AKT converted T cells towards regulatory phenotype and module post-infarction recovery, probably via reduced ROS production. 

The below are my concerns which authors may consider addressing:

1) Figure 1: Please provide single staining panel for both markers (i.e. PD-L1+AKT+DAPI). Please be careful with exposure and further processing of the images. Indicate scalebar.

2) Figure 3: B-D, LAD is major surgical procedure, therefore mice number representation is crucial. Please modify these graphs to 'scatter plots'

3) Figure 4: A. Poor quality images. Please provide with better images. B-same as (2)

4) Figure 5: Graphs are confusing. Why did authors combine Casp3 and NFκB? Please unify the size for all graphs.

5) Figure 6: Please provide a gating strategy

6) Since authors found increased CD25+ cells in the heart of AdMSC PD-L1 AKT group, Fig1 data can be validated by analyzing PD-L1 and AKT expression in heart immunofluorescence.

Authors should emphasize more convincingly how their findings supports their hypothesis by improving the Tregs role more detailed. 

Due to the poor images, graphs presentation and weak discussion, I am not in support of this manuscript at this level.

Round 2

Reviewer 1 Report

Comments and Suggestions for Authors

The MS is good. It can be accepted in present condition

Author Response

Thank you very much for the Reviewer.

Reviewer 2 Report

Comments and Suggestions for Authors

Comments:

Comments #5 to #9, not addressed satisfactorily. If further experiments are not feasible, authors should support the comments with appropriate references.

Reviewer 3 Report

Comments and Suggestions for Authors

I appreciate the authors for revising the manuscript. Authors response to my comments were shockingly unwelcome. Moreover, Some of my concerns were unaddressed.

1) I requested authors provide a single panel staining image with PD-L1+AKT+DAPI

2) Thanks for changing the graphs to 'scatter plots'

3) I requested that authors provide better quality images i.e. 'A'

4) Fig 6: Please label the 'y' axix with marker investigated. ie. B is casp3+ cells (% of total nuclei)

5) Why dint authors used CD45, viability markers?

6) Authors argument is invalid. Supply with data as requested.

7) Thanks for improving the discussion. 

Even though I conveyed them clearly, the authors ignored them. I urge authors to read my comments and respond accordingly.
